# Different types of theta rhythmicity are induced by social and fearful stimuli in a network associated with social memory

**Alex Tendler, Shlomo Wagner\***

Sagol Department of Neurobiology, University of Haifa, Haifa, Israel

**Abstract** Rhythmic activity in the theta range is thought to promote neuronal communication between brain regions. In this study, we performed chronic telemetric recordings in socially behaving rats to monitor electrophysiological activity in limbic brain regions linked to social behavior. Social encounters were associated with increased rhythmicity in the high theta range (7–10 Hz) that was proportional to the stimulus degree of novelty. This modulation of theta rhythmicity, which was specific for social stimuli, appeared to reflect a brain-state of social arousal. In contrast, the same network responded to a fearful stimulus by enhancement of rhythmicity in the low theta range (3–7 Hz). Moreover, theta rhythmicity showed different pattern of coherence between the distinct brain regions in response to social and fearful stimuli. We suggest that the two types of stimuli induce distinct arousal states that elicit different patterns of theta rhythmicity, which cause the same brain areas to communicate in different modes.

## Introduction

Oscillatory brain activity, mostly categorized to the theta (3–12 Hz), beta (12–30 Hz), and gamma (30–80 Hz) bands, is thought to coordinate neural activity in vast neuronal assemblies dispersed over different brain regions (*Buzsáki and Draguhn, 2004*). This type of coordination may underlie high level cognitive functions, such as speech and social communication (*Uhlhaas and Singer, 2006*; *Uhlhaas et al., 2009*) that are impaired in autism spectrum disorders (ASD) (*Geschwind, 2009*). Increasing evidence suggest that individuals with ASD show deficits in long-range neuronal communication associated with low-frequency rhythms, such as the theta rhythm (*Geschwind and Levitt, 2007*; *Rippon et al., 2007*; *Wass, 2011*). Nonetheless, a clear connection between rhythmic brain activity and social behavior has not yet been established.

Mammalian social organization depends on the ability to recognize and remember individual conspecifics (*Wiley, 2013*). This social recognition memory (SRM) can be assessed in rodents using their innate tendency to investigate novel conspecifics more persistently than familiar ones (*Gheusi et al., 1994*). In the SRM habituation–dishabituation test, social memory is assessed by the gradual reduction in the amount of time the animal spends investigating a particular social stimulus during consecutive encounters (*Ferguson et al., 2002*). This short-term memory was shown to be mediated mainly by chemical cues (semiochemicals) perceived via the main and accessory olfactory systems (*Dulac and Torello, 2003*). Upon binding of semiochemicals to the receptors expressed by the sensory neurons of the main olfactory epithelium and the vomeronasal organ, sensory information is conveyed to the main (MOB) and accessory (AOB) olfactory bulbs, respectively (*Dulac and Wagner, 2006*). Both bulbs then project, directly and indirectly, to the medial amygdala (MeA) (*Pro-Sistiaga et al., 2007*; *Kang et al., 2011*) that is thought to transfer the information to the hippocampus through the lateral septum (LS) (*Bielsky and Young, 2004*). The MOB projects also to several cortical areas comprising the primary olfactory cortex, of which the piriform cortex (Pir) is best characterized (*Wilson and Sullivan, 2011*) (*Figure 1*).

**\*For correspondence:**
shlomow@research.haifa.ac.il

**Competing interests:** The authors declare that no competing interests exist.

**eLife digest** For the brain to function correctly, the activities of multiple regions must be coordinated. This coordination is thought to be carried out by waves of electrical activity in the brain. One of the most prominent signals within these waves is called the theta rhythm.

The theta rhythm is thought to help coordinate neural activity between the regions of the brain that are involved in learning and memory. However, theta rhythms also appear when subjects encounter emotional stimuli, which suggests that they might have a role in social cognition. Consistent with this idea, theta rhythms are reduced in individuals with autism spectrum disorders, but the exact nature of the relationship between theta rhythms and social behavior has remained unclear.

Tendler and Wagner have now addressed this question directly by implanting electrodes into five brain regions that are active when rats engage in social interactions. Exposing a rat to a social stimulus, such as an unfamiliar visitor rat, caused the intensity of theta rhythms to increase in this network. This change was temporary, with the theta rhythms gradually returning to normal as the novelty of the visitor wore off.

An increase in the intensity of theta rhythms also occurred in the same network when the rats encountered a fearful stimulus, such as a tone that had previously signaled the delivery of a mild electric shock. Notably, however, the fearful stimulus led to an increase in low frequency theta rhythms, whereas the social stimulus led to an increase in high frequency theta rhythms.

These results suggest that social and fearful stimuli give rise to two different forms of alertness or arousal, which are reflected by the two types of theta rhythms in this network within the brain. Tendler and Wagner also suggest that the distinct frequencies of theta rhythms might be used to support different forms of communication between various regions of the brain, depending on the emotional value of the stimuli (for example, are they social or fearful stimuli?) encountered by the animal. This means that emotional states might be dictating cognitive processes such as learning and memory.

Here we hypothesized that social behavior is associated with an elevation of rhythmic activity in the network of brain areas that process social stimuli. To examine this hypothesis, we recorded electrophysiological activity from the brains of freely behaving adult male rats performing the SRM paradigm (*Video 1*). A telemetric system was used to record from wire electrodes chronically implanted in the five aforementioned brain regions: MOB, AOB, MeA, LS, and Pir (*Dulac and Wagner, 2006*). We found that social encounters were associated with enhancement of brain rhythmic activity, specifically at 7–10 Hz range, in all brain regions. This enhancement that was proportional to the degree of novelty of the social stimulus appeared to reflect an internal brain-state associated with social arousal. In contrast, a fear-conditioned tone, which is associated with fear arousal, induced rhythmicity in the low theta range (3–7 Hz) in the same network of brain regions. Moreover, social and fearful stimuli elicited different patterns of change in coherence between the distinct brain regions. We hypothesize that these two types of stimuli induce distinct arousal states in the animal, which are reflected by the different kinds of theta rhythmicity. We further suggest that the distinct types of theta rhythmicity support different modes of communication between the various brain areas. These in turn may modify cognitive processes such as memory acquisition and recall depending on the value and saliency of the stimulus by enhancing synchronous neuronal activity between remote neuronal assemblies.

## Results

### Brain theta rhythmicity is modulated by the novelty of the social stimulus

Electrophysiological recordings were carried out in the brains of freely behaving adult male rats performing the SRM habituation–dishabituation paradigm (*Figure 2A*). We first analyzed the dynamics of the local field potential (LFP) in the course of the behavioral paradigm. A highly rhythmic LFP was recorded in all brain areas during social encounters (*Figure 2B*). Power spectral density (PSD) analysis of the LFP showed a prominent peak at ~8 Hz, typical for the high theta band (*Buzsáki and Draguhn, 2004*), in all areas (*Figure 2C*). The value of this peak, termed theta power (TP), was very

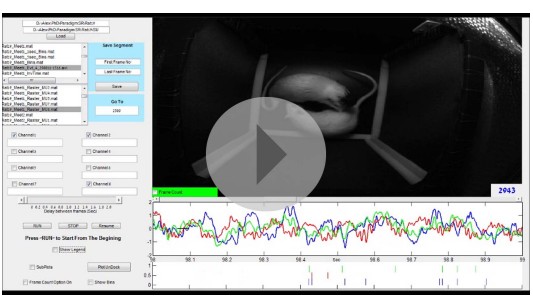

**Figure 1**. A simplistic scheme of sensory information flow in the network of brain regions thought to underlie social recognition memory. Social olfactory cues are detected by sensory neurons in the main olfactory epithelium (MOE) and vomeronasal organ (VNO). These neurons project to the main (MOB) and accessory (AOB) olfactory bulbs, which transmit information, either directly or indirectly (via the cortical nucleus of the amygdala—CoA) to the medial amygdala (MeA). The MOB also innervates the piriform cortex (Pir). The MeA projects to the lateral septum (LS), which innervates the hippocampus (Hip).

low in the absence of a social stimulus (Base, *Figure 2D–E*) but increased profoundly during the first encounter (Enc. 1). It then gradually decreased during further encounters with the same stimulus (Enc. 2–4), but increased again when another novel stimulus was introduced (Enc. 5). These changes in theta power during SRM testing closely followed the changes in investigation time (IT) (*Figure 2F*), with both parameters appearing to correlate with the degree of stimulus novelty.

We next analyzed the effect of social and non-social stimuli on the dynamics of investigation time and theta power in all recorded brain areas. As exemplified in *Figure 3A* (lower panel), exposure of an animal to either type of stimulus caused similar dynamics of the investigation time. However, there was a vast difference with regards to the theta power response to the social and non-social stimuli: whereas significant theta power modulation that was similar across all brain regions was observed with social stimuli, whether awake or anesthetized, object and odor stimuli did not cause such an effect (*Figure 3A*, upper panels).

Combined analyses of the modulation of both theta power and investigation time in animals exposed to social and object stimuli are presented in *Figure 3B*. Social stimuli caused a marked increase of mean theta power during Enc. 1 compared to Base, with the MOB and AOB showing the largest changes (6.2 dB/Hz) and other areas showing more moderate ones (4.0–5.1 dB/Hz). In all regions tested, the theta power decreased gradually during the habituation phase (Enc. 1–4) but returned the values obtained in Enc. 1 after dishabituation (Enc. 5) ($p < 0.005$ one-way repeated measures ANOVA, *$p_{corr} < 0.05$ post-hoc* paired t-test, *Figure 3—source data 1–2*). In contrast, object stimuli elicited a much weaker initial change from Base to Enc. 1 (1.1–2.7 dB/Hz) in all brain regions. Furthermore, the theta power showed modulation during the object paradigm similarly to the social paradigm only in the MOB, and even this change was not statistically significant ($p > 0.05$, *Figure 3—source data 1*). In a sharp contrast to the theta power, comparison of the investigation time of the social and object paradigms showed a highly similar course and magnitude of habituation and dishabituation that were statistically significant in both cases (*Figure 3C*, *Figure 3—source data 1–2*). Taken together, these results show that in almost all recorded brain areas, theta power is modulated by the degree of novelty of social but not object stimuli.

## The modulation of theta rhythmicity during social encounters is driven by an internal brain state of arousal

The lack of theta power modulation despite the clear investigation time modulation induced by object stimuli rejects the possibility that the theta rhythmicity is caused by the investigative behavior. We therefore reasoned that rather, theta power modulation may reflect processes that are either directly driven by the sensory input (Bottom-Up processes) or induced by an internal state of the brain that is modulated by the saliency of the social stimulus (Top-Down processes).

**Video 1.** Social encounter between two adult male rats in the experimental arena. The recorded subject carries a black transmitter equipped with a flashing led light on its head. Frame number is shown in the right low corner. The graph below the video shows the LFP recorded in the AOB (blue), MOB (red), and MeA (green). The bottom graph shows raster plots of spikes detected from the recorded multi-unit activity signal.

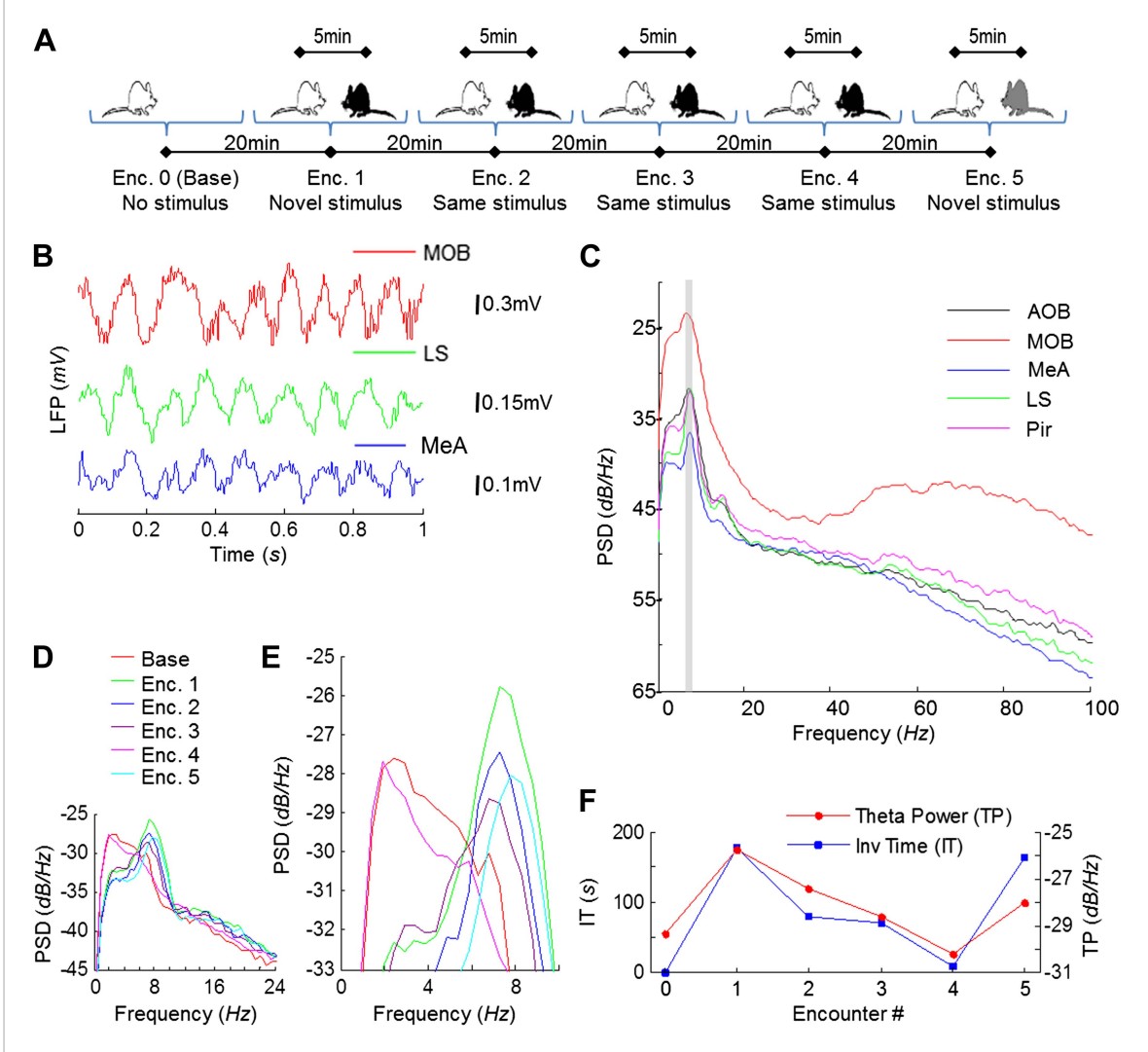

**Figure 2**. Theta rhythmicity in the rat brain is enhanced during social encounters, in correlation with the novelty of the social stimulus. (**A**) A scheme of the habituation–dishabituation SRM paradigm. (**B**) Examples of LFP traces recorded in the MOB, LS, and MeA during a social encounter. (**C**) Power spectral density (PSD) analyses of a 5-min LFP recording from all five brain areas during a social encounter. Gray bar represents the 7–9 Hz band. (**D**) Superimposed PSD analyses of LFP recordings from the MeA of one animal during the various stages of the SRM test. (**E**) As in **D**, zooming on the 4–10 Hz range. (**F**) The ~8 Hz PSD peak (TP) and social investigation time (IT) for the same experiment as in **D**, plotted as a function of the encounter number. Encounter 0 represents no stimulus (Base).

In order to distinguish between these two possibilities, we continued our recordings for 5 min after the stimulus was removed from the arena (Post 1–5). As depicted in *Figure 4A*, the theta rhythmicity did not cease with the removal of the social stimulus following Enc. 1, but remained at a high level during most of the Post 1 period (for spectrograms of the full experiment, see *Figure 4—figure supplements 1–5*). Plotting the mean instantaneous theta power as a function of time, revealed that this was true for all encounters with a social stimulus. In contrast, encounters with object stimuli were followed by a sharp drop in the theta power to a low level almost immediately following stimulus removal (*Figure 4B*, for all other brain areas see *Figure 4—figure supplements 6–7*). This significant reduction in mean theta power between the Enc. and Post periods of the object paradigm was characteristic of all brain areas (*Figure 4C*, *p < 0.05 paired t-test, *Figure 4—source data 1*). In contrast, high theta power levels were found in both these periods in the social paradigm (p > 0.05). Moreover, all encounters with social stimuli showed a steep but gradual increase in theta power during the first 15 s in which the stimulus was being

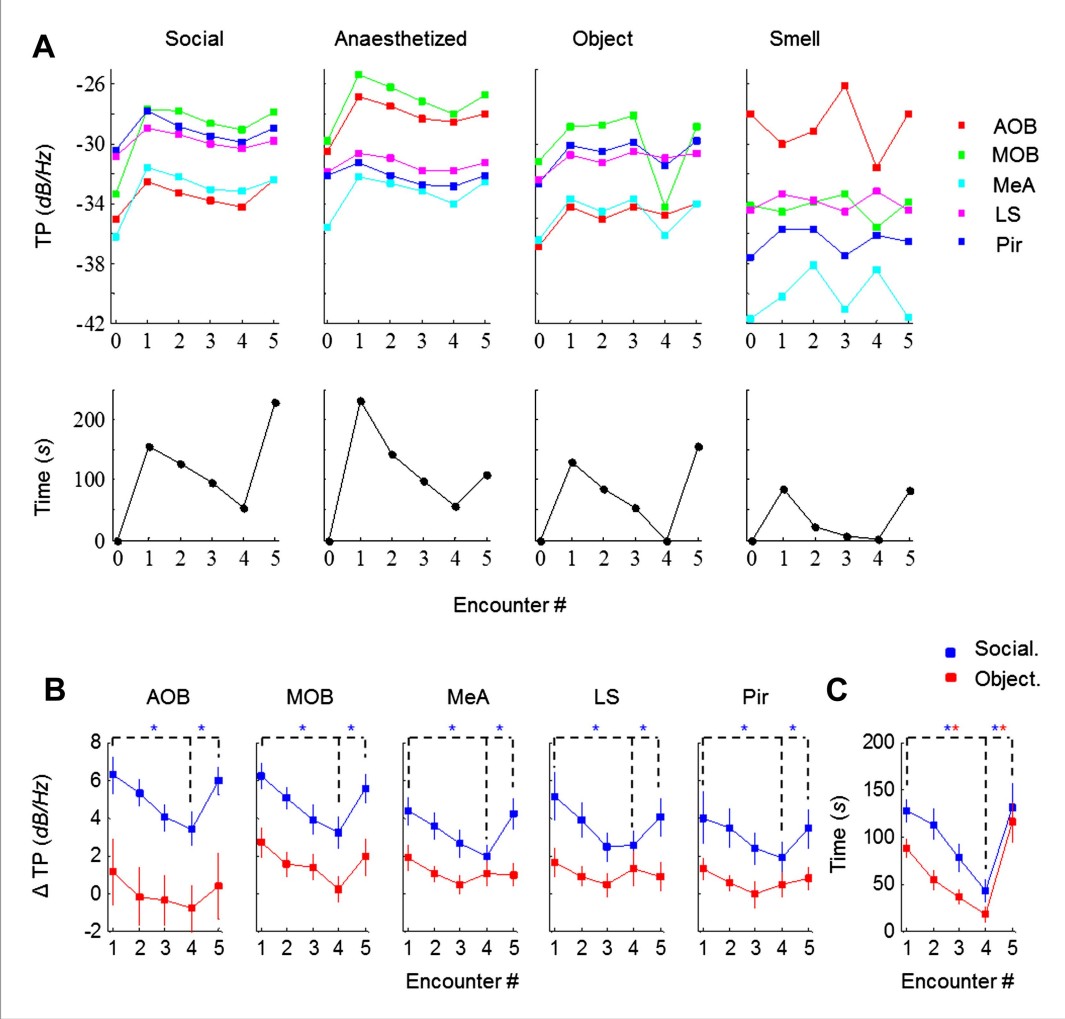

**Figure 3**. Theta rhythmicity is modulated by the novelty of social, but not other tested stimuli. (**A**) TP for all brain areas (upper) as well as IT (lower) during the SRM test of one animal, using awake and anesthetized social stimuli as well as object and smell stimuli, all except smell tested with the same animal. (**B**) Mean TP for the various brain regions averaged (±SEM) and plotted as a function of the test stage, for social (blue, n = 8) and object (red, n = 6) stimuli. A significant difference was found between the various encounters in all brain regions for social stimuli ($p < 0.005$, one-way repeated measures ANOVA, **Figure 3—source data 1A**), while no difference was found for object recognition ($p > 0.05$, **Figure 3—source data 1B**). *Post hoc* paired t-test showed significant differences between Enc. 1 and Enc. 4 as well as between Enc. 4 and Enc. 5 (dashed lines) in all brain regions for social stimuli (*$p_{corr} < 0.05$, **Figure 3—source data 2**). (**C**) As in **B**, for the IT of the social and object paradigms. Unlike the TP, both paradigms showed similarly significant modulation of the IT (**Figure 3—source data 1–2**).

The following source data are available for figure 3:

**Source data 1**. Theta power (TP) modulation between encounters.

**Source data 2**. Statistical assessment of habituation and dishabituation.

transferred into the arena (**Figure 4A,D**, gray bars). This rise in theta power probably reflects the subject's anticipation for a social meeting, as there was no similar increase with object stimuli (**Figure 4D**). Altogether, these data suggest that the changes in theta power during the SRM test reflect a graded internal brain state of arousal that is proportional to the novelty of the social stimulus and slowly fades away after its removal.

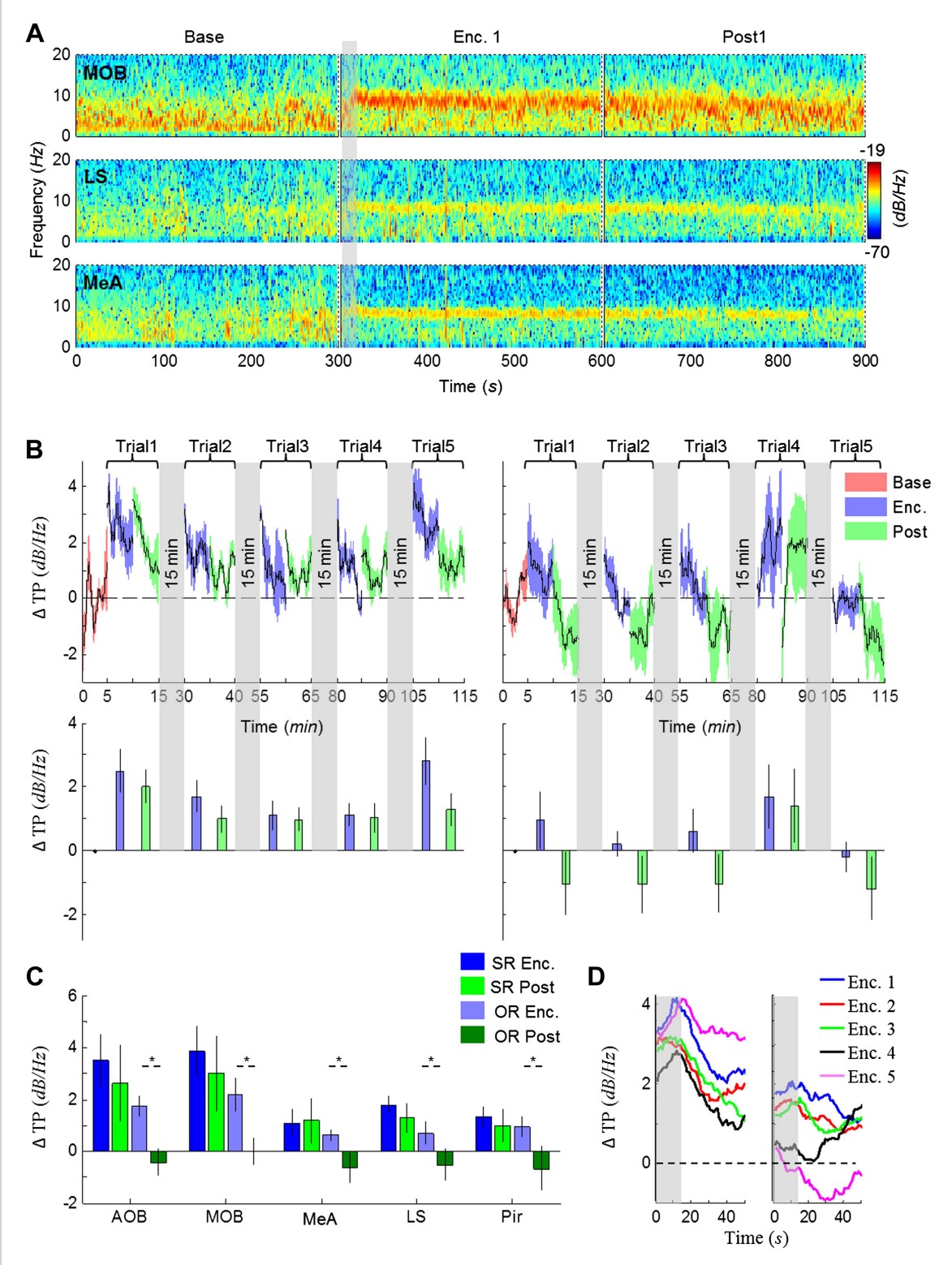

**Figure 4**. Modulation of the theta rhythmicity by social stimulus novelty reflects an internal state in the brain. (**A**) Color-coded spectrograms of the LFP recorded in the MOB (upper), LS (middle), and MeA (lower) for 5 min before (Base), during (Enc. 1), and after (Post 1) the first encounter of the SRM test. All spectrograms are averages of five animals (4 animals for LS). Gray bar marks the 15 s needed for stimulus transfer to the arena. (**B**) Upper—instantaneous ΔTP (change from mean Base) in the LS averaged over four rats (±SEM) during the Enc. and Post periods of all trials (1-5), for social (left, n = 5) and object (right, n = 4) paradigms. The 15-min breaks between last Post and next Enc. periods are labeled with gray bars. Lower—mean (±SEM) values for the

*Figure 4. continued on next page*

*Figure 4. Continued*

corresponding periods shown above. (**C**) Comparison of mean ΔTP averaged over all trials (1-5) for each brain area, between the Enc. and Post periods of the social and object paradigms (*$p < 0.05$, paired t-test, *Figure 4—source data 1*). (**D**) Left—the instantaneous ΔTP shown in **B**, expanded to show the initial 50 s of all encounters. Gray area represents the 15 s needed for stimulus transfer to the experimental arena. Right—The same for object stimuli.

The following source data and figure supplements are available for figure 4:

**Source data 1**. Comparison of ΔTP between Enc. and Post periods.

**Figure supplement 1**. Mean LFP spectrograms across the SRM paradigm for the AOB.

**Figure supplement 2**. Mean LFP spectrograms across the SRM paradigm for the MOB.

**Figure supplement 3**. Mean LFP spectrograms across the SRM paradigm for the MEA.

**Figure supplement 4**. Mean LFP spectrograms across the SRM paradigm for the LS.

**Figure supplement 5**. Mean LFP spectrograms across the SRM paradigm for the Pir.

**Figure supplement 6**. Comparison of mean instantaneous TP between social and object stimuli, for the AOB and MOB.

**Figure supplement 7**. Comparison of mean instantaneous TP between social and object stimuli, for the MeA and Pir.

## The theta rhythmicity during social behavior emerges from multiple sources with dynamic coherence between brain areas

The theta rhythmicity recorded in the network may reflect a single rhythm originating from one source. In that case, the various brain regions are expected to display high correlation and similar dynamics of coherence in their rhythmicity. Alternatively, if it represents a combination of multiple independent rhythms arising from several sources, we expect low correlation and differential dynamics of coherence between various brain regions. To discriminate between these possibilities, we first examined the cross-correlation of the LFP, filtered in the theta range, between the MeA and the other brain areas. Despite the fact that both areas are directly connected to the MeA, the strongest correlation appeared with the LS and the weakest with the MOB (*Figure 5A–D*). Moreover, whereas the correlation between the MeA and LS was significantly higher during Enc. 1 (blue) compared to Base (red), the MOB showed consistently low correlation with the MeA during both periods. The presence of a social stimulus thus appears to differentially affect the correlation of theta rhythmicity between distinct brain areas.

We next analyzed the coherence of the LFP signal among all brain areas during the Base, Enc. 1, and Post 1 periods of the SRM paradigm. As depicted in *Figure 6A*, the coherence between the MeA and the LS showed several prominent peaks, especially in the theta and gamma bands. Yet, while no change was recorded in the gamma band, the theta coherence showed a significant increase between the Base and Enc. 1. Furthermore, similarly to theta rhythmicity itself (*Figure 4*), the high coherence at theta range persisted during the Post 1 period despite the lack of a social stimulus (*Figure 6A,C*). In contrast, the coherence in theta band between the MeA and MOB remained low throughout all periods (*Figure 6B,C*). Analyses across all regions revealed a hierarchy in the theta coherence between the MeA and all other areas, ranging from a low level with the MOB and AOB, medium coherence with the Pir and high coherence with the LS (*Figure 6D*). This notion of functional hierarchy between brain regions is strengthened by the fact that despite their largest physical distance, the highest level of theta coherence was found between the MeAs in the two hemispheres (*Figure 6—figure supplements 1,3*). Furthermore, the theta coherence between the MeA and the higher brain centers (Pir, LS) significantly increased during Enc. 1 and Post 1 (*$p_{corr} < 0.05$, paired t-test, *Figure 6—source data 1*), while no change was recorded between the MeA and both areas of the olfactory bulb (MOB, AOB, $p_{corr} > 0.05$). This suggests the existence of at least two independent

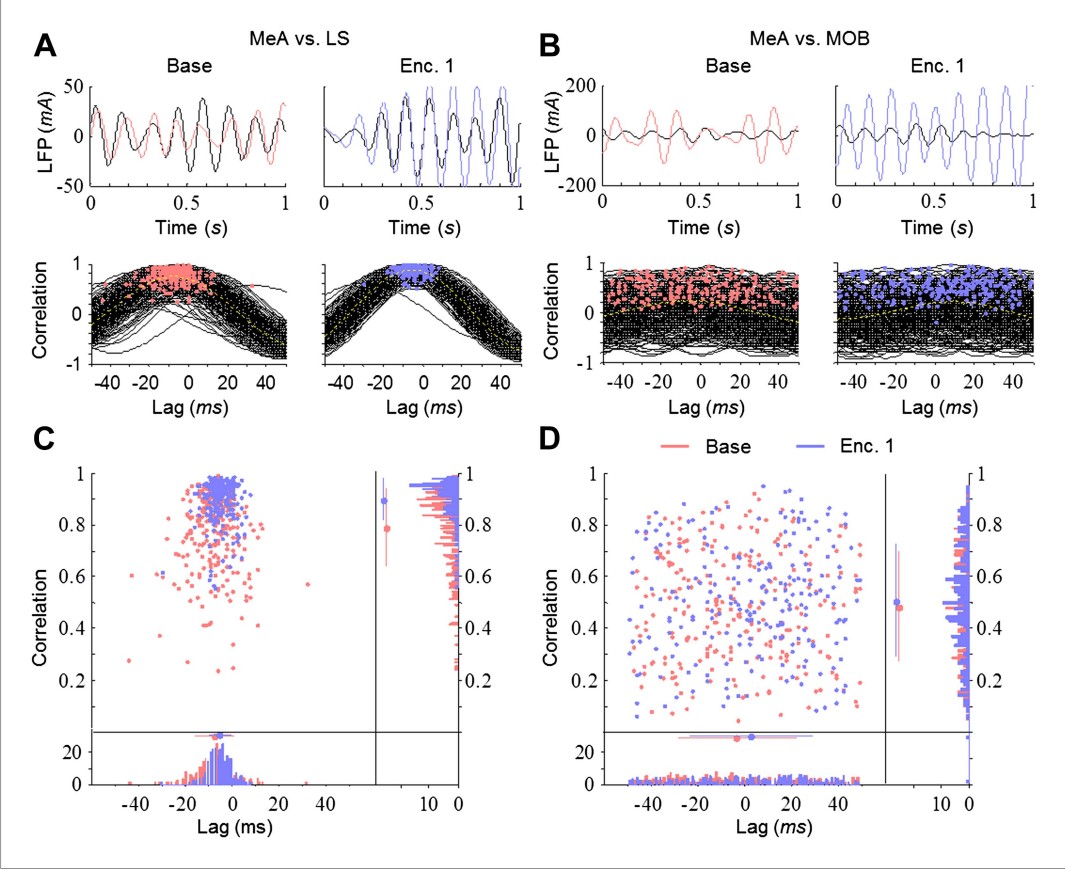

**Figure 5**. Differential and dynamic correlation of theta rhythmicity between specific brain regions. (**A**) Upper—superimposed LFP traces (filtered 5–11 Hz) from the MeA (black) and LS (colored) of one animal during Base (left, red) and Enc. 1 (right, blue). Lower—cross-correlations between both regions for each of the 300 s recorded during the same periods, with peaks labeled by colored dots. (**B**) Same as **A** for the MeA and MOB. (**C**) Middle—distribution of the cross-correlation peaks for the data in **A**. Borders—histograms of the cross-correlation peaks in the correlation (right) and lag (bottom) axes. Mean ± SD are marked to the left (correlation) or above (lag) the histograms. (**D**) Same as **C** for the data in **B**.

theta rhythms, one that governs the olfactory bulb and another that dominates higher brain structures. This conclusion is further supported by the findings that the MOB shows opposite relationships with all other brain areas; high coherence with the AOB and low coherence with the higher areas (*Figure 6E*, *Figure 6—figure supplements 2,3*). Moreover, a significant enhancement in theta coherence with the AOB was observed during Enc. 1 and Post1 (*$p_{corr} < 0.05$, paired t-test, *Figure 6—source data 1*), while all other regions showed no change ($p_{corr} > 0.05$, paired t-test). Interestingly, similar enhancement of theta coherence between the AOB and MOB was recorded with object stimuli, while these stimuli did not cause any enhancement of the coherence between the MeA and LS or Pir (*Figure 6F,G*, *Figure 6—source data 1*). Together, these data support multiple sources of theta rhythmicity in the network.

## Distinct types of theta rhythmicity are induced in the same brain regions by social and fearful stimuli

Theta rhythmicity was previously found to be elicited in several brain regions during states of arousal, mainly in response to fearful stimuli (*Knyazev, 2007*). This phenomenon was best studied in the context of fear learning in a network of brain regions comprising the basolateral complex of the amygdala (lateral and basolateral amygdala), hippocampus, and medial prefrontal cortex (*Pape and Pare, 2010*). In this network, a recall of a fearful memory, induced by a fear-conditioned stimulus, elicits robust theta rhythmicity that shows high coherence between these brain regions (*Paré and Collins, 2000*;

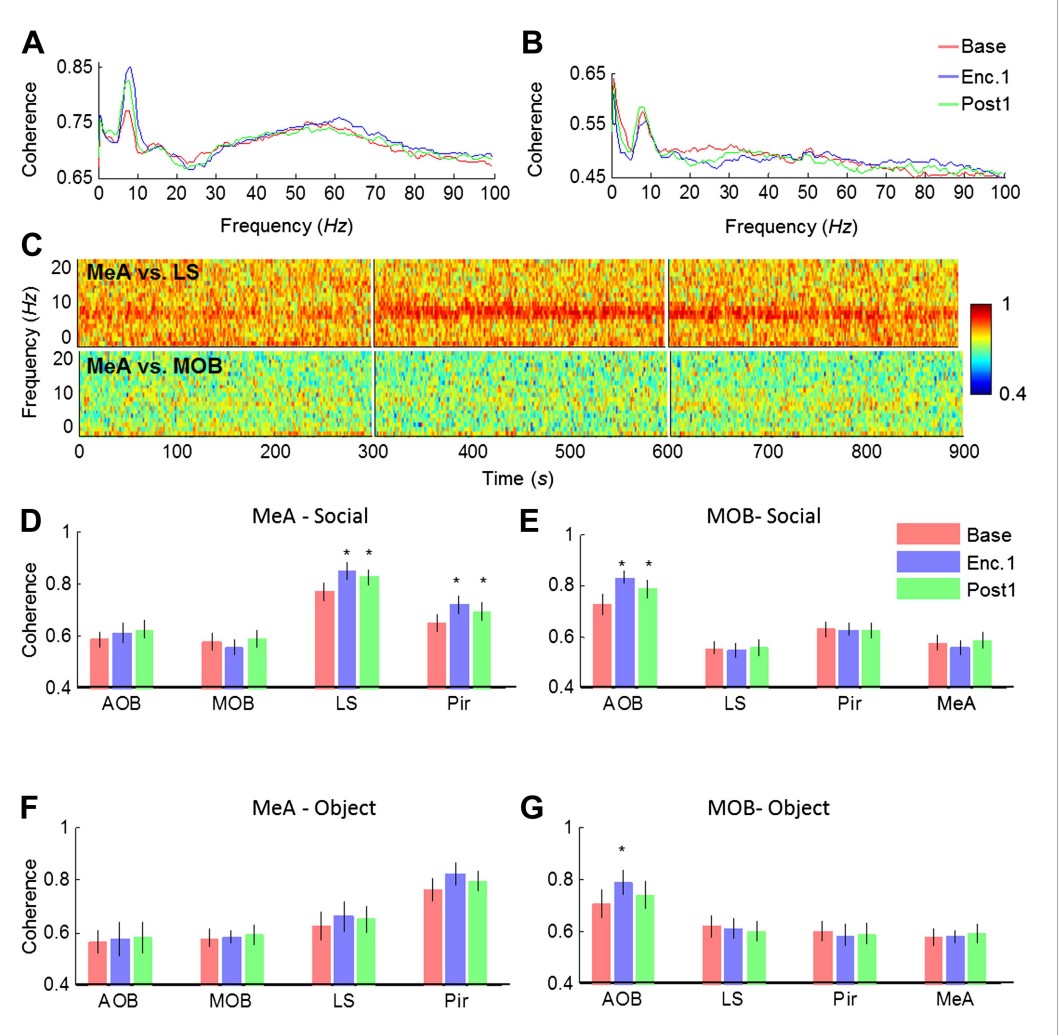

**Figure 6**. Theta coherence between specific brain regions increases during social encounter. (**A**) Mean (n = 10 animals) coherence (0–100 Hz) of the LFP signals recorded in the MeA and LS during Base, Enc. 1, and Post 1 periods. (**B**) Same animals, coherence analysis between the MeA and MOB. (**C**) Spectrograms (0–20 Hz) of the coherence analyses shown in **A** (between MeA and LS, upper panel) and **B** (between MeA and MOB, lower panel). (**D**) Mean coherence at 8 Hz between the MeA and all other areas (MOB, AOB n = 11; LS, Pir n = 10) during the Base, Enc. 1, and Post 1 periods of social encounter (*$p_{corr}$ < 0.05, paired t-test, *Figure 6—source data 1A*). (**E**) Same as **D**, for coherence of the MOB with all other areas (*$p_{corr}$ < 0.05, paired t-test, *Figure 6—source data 1B*). (**F**) Same as **D**, for object stimuli (*Figure 6—source data 1C*). (**G**) Same as **E**, for object stimuli (*$p_{corr}$ < 0.05, paired t-test, *Figure 6—source data 1D*).

The following source data and figure supplements are available for figure 6:

**Source data 1**. Assessment of change in theta Coherence from Base to either Enc. 1 or Post 1.

**Figure supplement 1**. Mean spectrograms of coherence between the MeA and all other areas during trial 1 of the SRM paradigm.

**Figure supplement 2**. Mean spectrograms of coherence between the MOB and all other areas during trial 1 of the SRM paradigm.

**Figure supplement 3**. Mean theta coherence during trial 1 of the SRM paradigm.

*Pare et al., 2002*; *Seidenbecher et al., 2003*; *Pape et al., 2005*; *Popa et al., 2010*). Here, we examined whether the brain state-induced theta rhythmicity during the SRM paradigm is similar to the fear-induced rhythmicity. To address this question, we compared the theta rhythmicity induced by a social encounter to that of a fear stimulus within the social network that we investigated. To that end, a new cohort of six animals was implanted with wire electrodes as before, with an additional electrode in the nucleus accumbens (NAcc), which was recently shown to be involved in social motivation (*Dölen et al., 2013*; *Gunaydin et al., 2014*). These animals were fear-conditioned by coupling a 40-s long tone to an electrical foot shock for five consecutive times separated by 180-s intervals (*Figure 7—figure supplement 1A*). A day later the electrical activity was recorded in two consecutive sessions, each following a 30 min of habituation to the arena. The first session was recorded during a recall of fear memory (FC experiment), and the second during a 5-min long encounter with a novel social stimulus (SR experiment). During the FC experiments (*Figure 7—figure supplement 1B*), introduction of the fear-conditioned tone caused animals to begin moving intensively, followed by immobility (freezing) towards the end of the tone, in anticipation of the foot shock. The freezing response was especially significant at the end of the first tone (*Figure 7—figure supplement 1C*). Thus, the fear-conditioned tone caused a robust arousal state that was associated with intense movement of the conditioned animals. We then compared the theta rhythmicity between the FC and SR experiments. A PSD analysis of the LFP signals recorded in the LS during 5 min prior to stimulus introduction (Base) yielded a similar profile in both cases (*Figure 7A*, red). However, the PSD was very different between the two types of stimuli during the first 15 s following stimulus introduction (Stimulus) (*Figure 7A*, blue). Whereas the fear stimulus caused a marked peak at the low theta range (3–7 Hz), the social stimulus resulted in a peak at the high theta range (7–10 Hz). This change is clearly observed when subtracting the Base PSD from the Stimulus profile (*Figure 7B*). These differences appeared in all recorded brain regions (*Figure 7C*) and Statistical analysis showed a highly significant interaction between the type of experiment (FC or SR) and theta band (*Figure 7D*) (**$p < 0.01$, two-way repeated measures ANOVA, *Figure 7—source data 1*). Thus, we conclude that fearful and social stimuli cause changes in very different ranges of theta rhythmicity in the same limbic network of brain regions. We suggest that these different types of theta rhythmicity reflect distinct arousal states; the low theta reflects aversive arousal that is associated with fear while the high theta reflects appetitive arousal associated with a social encounter.

## Distinct changes in coherence are induced in the network by social and fearful stimuli

We next examined how the coherence between the various brain regions changes in response to the two types of arousing stimuli. *Figure 8A* depicts the coherence between the MeA and LS during Base and Stimulus periods of FC and SR experiments, respectively. The change in coherence of the two stimuli is presented in *Figure 8B* and reveals a positive peak at the high theta range for the social encounter, and at the low theta range for the fear memory recall. A quantitative analysis of all coherence changes within the network in both ranges showed that this tendency generally holds for all pairs of brain regions (*Figure 8C*). Accordingly, most pairs showed a statistically significant interaction between the type of experiment (FC or SR) and theta band (high or low) (*$p < 0.05$, **$p < 0.01$, two-way repeated measures ANOVA, *Figure 8—source data 1*). Nevertheless, the magnitude of changes was different between distinct pairs. For example, the changes in the coherence between the LS and NAcc were much smaller than those recorded between the Pir and NAcc and did not show any statistical significance. Moreover, the increases of coherence between the AOB-MOB and MOB-Pir pairs were much bigger in SR compared to the FC experiment. We conclude that the distinct arousal states are characterized by distinct patterns of coherence changes within that same network of brain regions (*Figure 9*).

## Discussion

This study demonstrates that an encounter with a social stimulus causes increased LFP rhythmicity in the high theta range (7–10 Hz), in a network of limbic brain areas associated with social memory. Strikingly, the change in theta rhythmicity is directly proportional to the novelty of the social partner, and may thus be considered a neuronal correlate of short-term social memory (*Liebe et al., 2012*). Since the modulation of theta rhythmicity is observed even when anesthetized stimuli are used, we infer that it does not depend on the behavior of the social stimulus. Despite the similarity in investigative behavior, such modulation of theta rhythmicity is not observed with object stimuli,

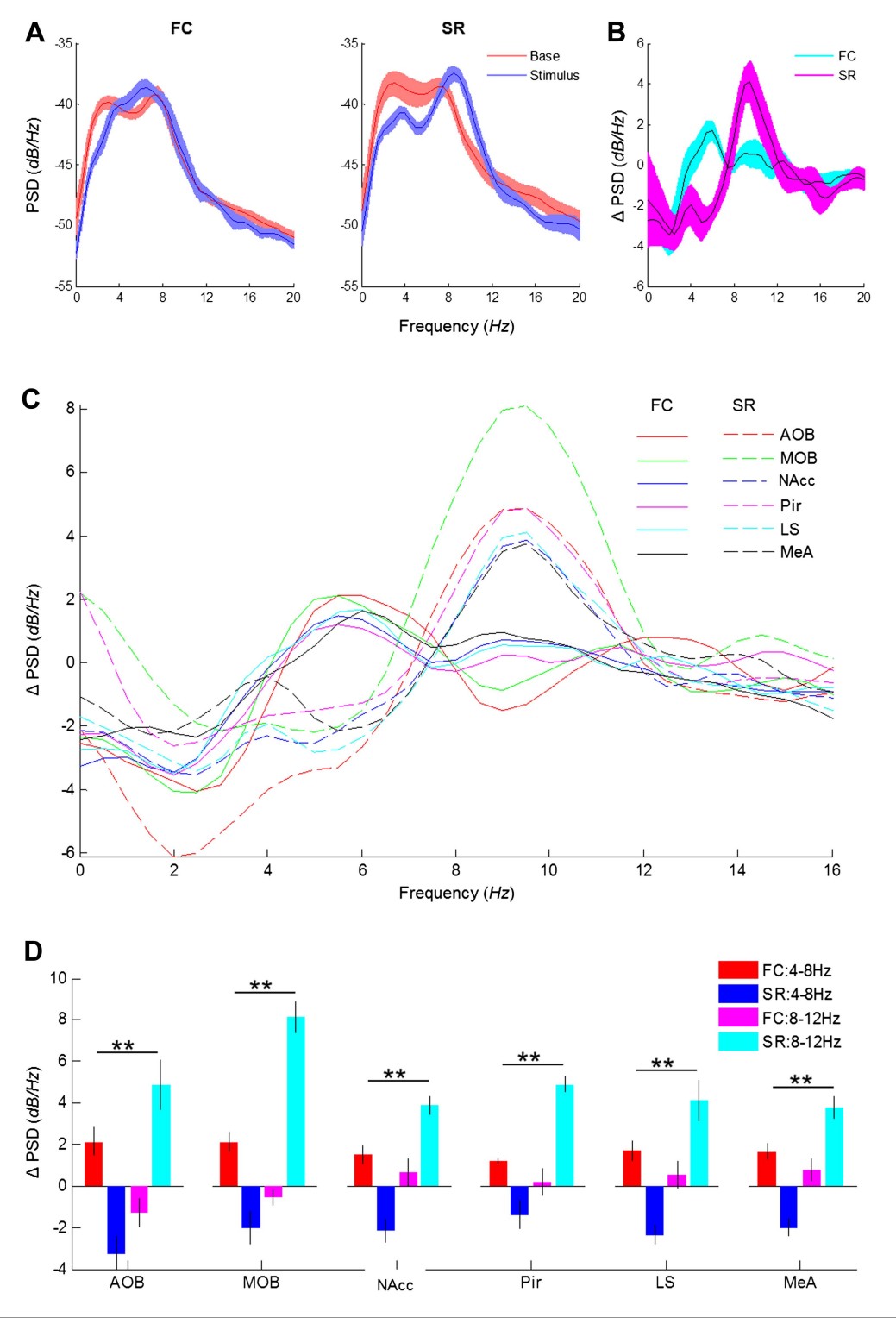

**Figure 7**. Distinct types of theta rhythmicity are induced by social and fearful stimuli. (**A**) PSD analyses (0–20 Hz) of LFP signal recorded in the LS of one animal, 5 min prior to stimulus introduction (Base, red) and 15 s following it (Stimulus, blue) during fear memory recall (left, FC) and social encounter (right, SR). (**B**) The change between Stimulus and Base PSD analyses (Stimulus minus Base) shown in **A**, for FC and SR, superimposed. (**C**) Mean change in PSD profile for all brain areas of the same six animals during the FC (continuous lines) and SR (dashed lines) experiments. (**D**) Mean (±SEM) values of the peak change in PSD in the low (4–8 Hz, red and blue) and high (8–12 Hz,
*Figure 7. continued on next page*

*Figure 7. Continued*

pink and light blue) theta ranges for the FC (red and pink) and SR (blue and light blue) experiments (**$p < 0.01$, experiment X theta range interaction, two-way repeated measures ANOVA, *Figure 7—source data 1*).

The following source data and figure supplements are available for figure 7:

**Source data 1**. Comparison of change in theta power in low and high theta bands between social and fearful stimuli.

**Figure supplement 1**. Arousal-driven locomotion during recall of fear memory.

suggesting that it is social specific. Since the augmented theta rhythmicity and the associated increase in theta coherence persist beyond the removal of the social stimulus itself, we conclude that these parameters do not mirror sensory inputs but rather reflect a state of arousal that slowly fades away. This is in agreement with the fact that the increase in theta power occurs prior to the actual introduction of the social stimulus in the arena, suggesting increased arousal due to the anticipated social encounter. Finally, since the change in theta rhythmicity during the SRM test correlates with the novelty of the social stimulus, we posit that it reflects a graded level of arousal, which is proportional to the stimulus saliency.

One of the questions that arise from the study is whether the social encounter-induced state of arousal is elicited by the 'social' quality of the stimulus or whether it simply results from the complexity of the stimulus. Notably, the social stimulus is much more complex than the single object or odor stimuli that we used as controls. It emits a complex mixture of odors and semiochemicals, and in addition to the main and accessory olfactory systems, it also stimulates the visual, auditory, and somatosensory systems. It is not likely that the full complexity of the social stimulus may be mimicked by the use of any artificial mixture of odors, hence the possibility that the arousal state results from the complexity of the stimulus cannot be excluded. On the other hand, at least as regards to fear-associated arousal, it is well documented (*Takahashi et al., 2008*) that a very simple cue is sufficient to evoke a state of arousal, such that is observed by the freezing of rodents in response to the pure odorant 2,3,5-Trimethyl-3-thiazoline (TMT), a component of fox odor (*Fendt et al., 2003*), or to a pure tone in a fear conditioning paradigm (*Rogan et al., 1997*). This suggests that the factor that determines the state of arousal is not the complexity of the stimulus but rather the information it embodies with regards to the natural environment of the animal.

Many studies, both in animals and humans, have linked brain theta rhythmicity to the processing of emotional cues (*Sainsbury and Montoya, 1984*; *Sainsbury et al., 1987a*; *Sainsbury et al., 1987b*; *Aftanas et al., 2001*; *Balconi and Pozzoli, 2009*; *Knyazev et al., 2009*; *Maratos et al., 2009*; *Luo et al., 2013*). In animals theta rhythmicity was mostly studied in the hippocampus (*Buzsáki, 2002*), where it was classified into two types, Type 1 and Type 2. The atropine-insensitive Type 1 theta rhythmicity shows higher frequency (8–12 Hz) and is thought to be associated mainly with voluntary movement. In contrast, atropine-sensitive Type 2 rhythmicity is characterized by lower frequency (4–8 Hz) and is thought to be linked to arousal during states of immobility (*Bland, 1986*; *Sainsbury, 1998*). Notably, Type 2 rhythmicity was mostly studied using states of fear and aversive stimuli and was shown to be induced by neutral stimuli if conditioned by fear or introduced in the presence of predators (*Sainsbury and Montoya, 1984*; *Sainsbury et al., 1987a*; *Sainsbury et al., 1987b*). The relationship of the two types of hippocampal theta rhythmicity and similar rhythms recorded from other brain regions, such as in our case, should be cautiously examined for several reasons. First, recent studies showed that in the hippocampus itself there are differences in the profile of theta rhythmicity between the earlier studied dorsal hippocampus and the more recently studied ventral hippocampus (*Adhikari et al., 2010*), the latter of which shows theta rhythmicity with stronger association to the one recoded in the mPFC (*Jacinto et al., 2013*), and may be dissociated from the dorsal hippocampus under certain conditions such as decision making (*Schmidt et al., 2013*). Second, even for the dorsal hippocampus the dichotomy between the two types of theta rhythmicity is far from being perfect with Type 2 rhythmicity reported to reach 12 Hz at some states and Type 1 rhythmicity reported to disappear during certain movements (*Sainsbury, 1998*). Interestingly, researchers reported that in cats the correlation between movement and Type 1 rhythmicity was good at the beginning of the experiments, when a lot of exploratory and object manipulation behavior was

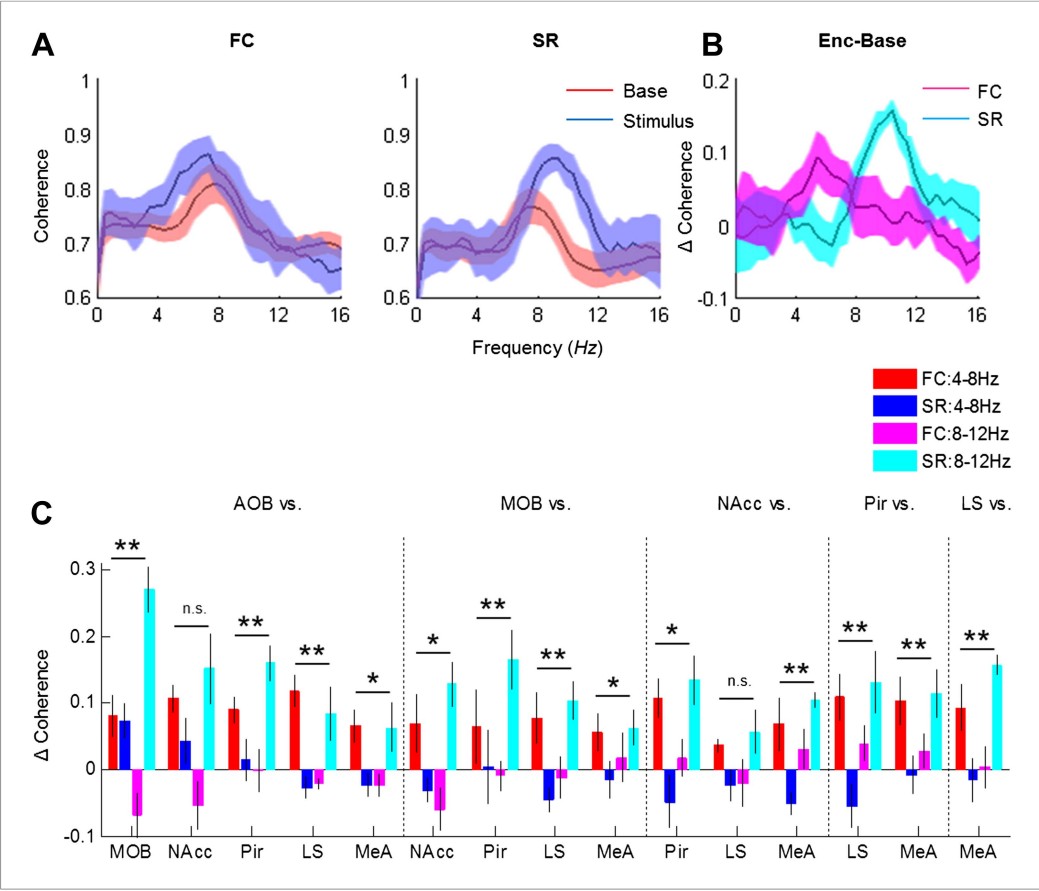

**Figure 8**. Distinct changes in theta coherence between various brain regions are induced by social and fearful stimuli. (**A**) Coherence analyses (0–20 Hz) of LFP signal recorded in the LS and MeA of one animal, 5 min prior to stimulus introduction (Base, red) and 15 s following it (Stimulus, blue) during fear memory recall (left, FC) and social encounter (right, SR). (**B**) The change between Stimulus and Base coherence analyses (Stimulus minus Base) shown in **A**, for FC and SR, superimposed. (**C**) Mean (±SEM) values of the peak change in coherence between all possible couples of brain areas in the low (4–8 Hz, red and blue) and high (8–12 Hz, pink and light blue) theta ranges for the FC (red and pink) and SR (blue and light blue) experiments (*$p < 0.05$, **$p < 0.01$, experiment X theta range interaction, two-way repeated measures ANOVA, *Figure 8—source data 1*).

The following source data is available for figure 8:

**Source data 1**. Comparison of change in coherence in low and high theta bands between social and fearful stimuli.

observed, but deteriorated towards the end of the experiments, when the animals were still moving but were uninterested in the task (*Sainsbury, 1998*). This might suggest that in the hippocampus too, high frequency Type 1 theta may be associated with sensory information processing during 'positive' arousal states associated with motivational voluntary movements, such as exploration, while low frequency Type 2 theta may be linked to 'negative' arousal states, such as those caused by fear, which is usually associated with freezing.

Regardless of the nature of hippocampal theta oscillations, theta rhythmicity associated with emotional states was reported in several other brain areas (*Bland et al., 1993*; *Bland and Oddie, 2001*; *Pignatelli et al., 2012*). Of particular interest is the finding that theta rhythmicity in a limbic network that includes the hippocampus, medial prefrontal cortex and the basolateral complex of the amygdala (lateral and basolateral amygdala) is associated with fear memories. Importantly, the consolidation and recall of long-term fear memory was found to be associated with elevated coherence of the theta rhythmicity in this network (*Paré and Collins, 2000*; *Seidenbecher et al., 2003*; *Pape et al., 2005*; *Popa et al., 2010*; *Lesting et al., 2013*), while its extinction was associated

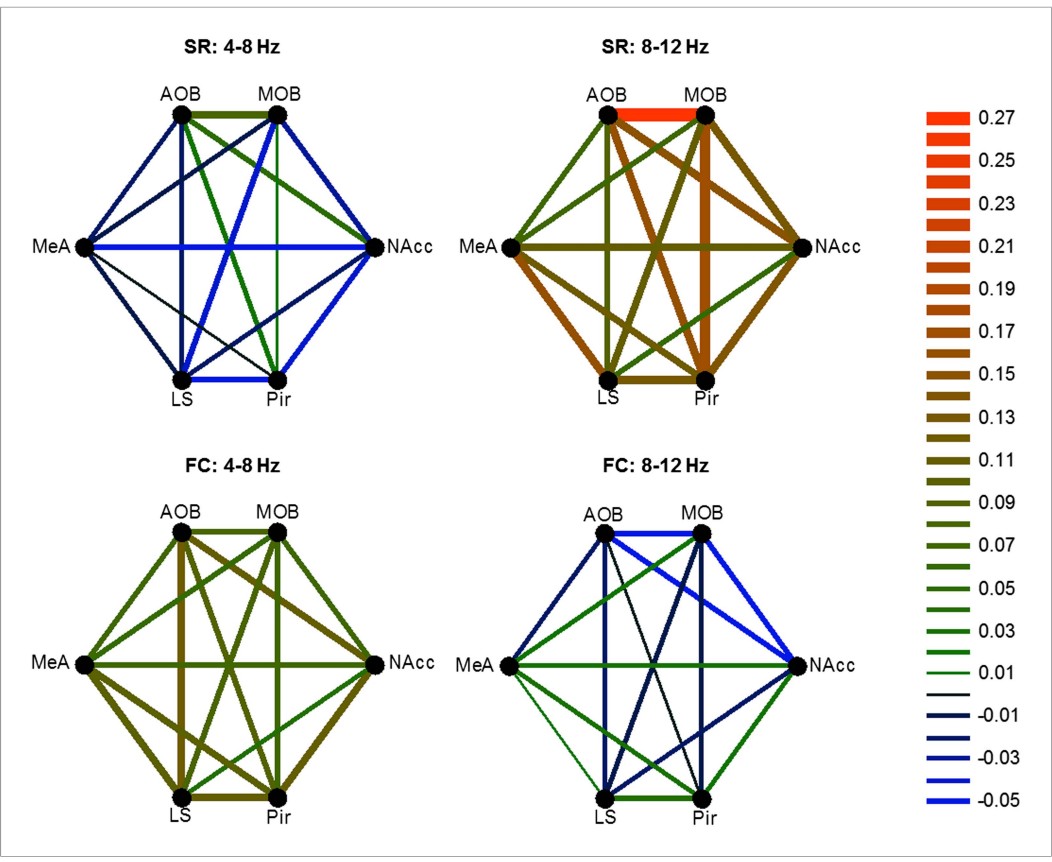

**Figure 9**. Different patterns of coherence change characterize the distinct arousal states. Graphical color-coded presentation of the mean changes in coherence for the FC and SR experiments.

with a decline in coherence, in a brain-region dependent manner (*Narayanan et al., 2007*). Moreover, interfering with theta coherence through local electrical micro-stimulation affected fear-memory recall and extinction depending on theta phase (*Lesting et al., 2013*). Thus, coordinated arousal-induced theta rhythmicity within this network seems to be involved in consolidation and recall of aversive memories (*Popa et al., 2010*; *Lesting et al., 2013*). Here, we demonstrated for the first time that similar phenomena occur in a distinct network of limbic areas that are linked to social memory, in the course of social encounters. Importantly, a comparison of the theta activity between social and fearful stimuli revealed that although both cause a state of arousal, the patterns of theta rhythmicity and coherence within the same network are completely different. First, in agreement with previous studies (*Paré and Collins, 2000*; *Seidenbecher et al., 2003*; *Pape et al., 2005*; *Popa et al., 2010*; *Lesting et al., 2013*), the recall of fear memory causes rhythmicity in the low theta range, while a social encounter elicits rhythmic activity in the high theta range. This suggests the existence of two types of arousal: fear-associated arousal and social-related arousal. Second, each of these conditions caused a distinct pattern of coherence changes between the same regions of the network. Given these results, we hypothesize that the distinct types of theta rhythmicity promote different communication protocols (*Kepecs et al., 2006*) for the coordination of neural activity in the network, which depends on the emotional state of the animal. Our results are in agreement with the hypothesis that theta rhythmicity facilitates cognitive processes such as memory formation that are associated with emotionally salient stimuli (*Pelletier and Paré, 2004*).

The source and distribution of theta rhythms in the mammalian brain are not fully understood (*Pignatelli et al., 2012*). This issue was extensively studied in the hippocampus (*Buzsáki, 2002*), which was shown to have the capacity to self-generate theta rhythmicity (*Goutagny et al., 2009*). Yet, as described above theta rhythmicity also exists in various cortical and limbic areas, where it shows dynamic coherence with the hippocampal theta rhythm. One area shown to display robust theta

rhythmicity is the olfactory bulb, where it follows the rhythm of respiration ('sniff cycle') (*Rojas-Libano et al., 2014*). Sniffing, similarly to whisking, is a sensory sampling activity, the rate of which dynamically changes throughout the theta band and is strongly influenced by internal arousal and motivational state of the animal (*Clarke and Trowill, 1971*; *Chang, 1992*). Specifically, high-frequency sniffing (8–12 Hz) develops in anticipation of reward delivery (*Freeman et al., 1983*; *Monod et al., 1989*; *Kepecs et al., 2007*; *Wesson et al., 2008*). The olfactory bulb theta rhythm and sniffing are not usually coherent with the hippocampal rhythm. However, in some odor-based learning tasks these rhythms do become transiently coherent (*Macrides et al., 1982*; *Kay, 2005*; *Martin et al., 2007*), a process that was suggested to be mediated by cholinergic neurons in the medial septum (*Tsanov et al., 2014*). Interestingly, whisking was shown to get occasionally phase locked with the sniff cycle (*Cao et al., 2012*; *Ranade et al., 2013*) or with the hippocampal theta rhythm (*Komisaruk, 1970*) during exploratory behavior. Thus, various generators of theta rhythmicity in the brain, such as those reflected by sniffing, whisking, or the hippocampal theta rhythm may become dynamically coupled by the brain neuromodulatory systems. While we did not monitor sniffing in our experiments, several recent studies reported changes in sniffing during both social interactions (*Assini et al., 2013*; *Wesson, 2013*) and fear conditioning (*Shionoya et al., 2013*). These studies showed that the sniff cycle adopt high-range theta rhythmicity during social interactions and low-range rhythmicity during fear conditioning. These differences are probably reflected by the distinct rates of theta rhythmicity that we record in the MOB and AOB during these conditions. This may explain our observation of high coherence between MOB-AOB and the low coherence each of them display with all other regions. Moreover, while the coherence between the MOB-AOB is increased during exploration of both social and object stimuli, the coherence between the LS–MeA increases only during social interactions. Thus, the theta rhythmicity displayed by the AOB and MOB probably emerges from a distinct generator, most likely the sniff cycle, that is separate from the one causing rhythmicity in higher brain areas. Furthermore, the significant differences in correlation and coherence dynamics between the various limbic areas suggest the involvement of distinct generators as well. For example, neither paradigm showed significant coherence changes between the LS-NAcc, as opposed to a significant increase in coherence between the LS-MeA or LS-Pir during social interactions. It should be noted that these differences cannot not be accounted for by local diffusion of LFP signals, since the LS is much closer to the NAcc than to the MeA or Pir. Direct synaptic connections cannot explain these differences either as the MeA shows very low coherence with the AOB, despite the strong bidirectional connections between them, but rather displays the highest coherence with the contralateral MeA, despite the lack of direct synaptic pathway (*Canteras et al., 1995*). Therefore, the differential coherence changes between distinct pairs of brain regions during the various conditions are most likely mediated by either a common input to these regions or via brain-region specific neuromudulatory systems. However, the arousal-driven modulation of theta rhythmicity which seems to be common to all brain regions is probably mediated by a general, brain-wide neuromodulatory mechanism such as neurohormonal activity (*Lee and Dan, 2012*; *Marder, 2012*).

An ever growing body of evidence implies rhythmic brain activity in various cognitive processes, particularly in memory acquisition and recall (*Fell and Axmacher, 2011*; *Buzsáki and Watson, 2012*; *Cannon et al., 2014*). Specifically, slow frequency rhythms such as the theta rhythm, are hypothesized to mediate communication between brain regions and to promote the temporal binding of neural assemblies in these areas into coherent networks subserving specific cognitive processes (*Buzsáki and Draguhn, 2004*; *Jutras and Buffalo, 2010*; *Benchenane et al., 2011*; *Fell and Axmacher, 2011*). During the last decade, several prominent theories implied a disordered or weak communication among brain regions as a major deficit underlying ASD etiology and symptoms (*Brock et al., 2002*; *Uhlhaas and Singer, 2006*; *Geschwind and Levitt, 2007*; *Kana et al., 2011*; *Wass, 2011*). Indeed, multiple recent studies found reduction in the power and coherence of slow brain rhythms, such as the alpha and theta rhythms, in ASD individuals (*Murias et al., 2007*; *Coben et al., 2008*; *Isler et al., 2010*; *Barttfeld et al., 2013*; *Doesburg et al., 2013*; *Machado et al., 2013*; *Kikuchi et al., 2015*). In agreement with these findings, our results suggest that arousal-driven theta rhythmicity may help bind correlated neuronal assemblies in distinct brain areas participating in cognitive and emotional processes underlying social behavior. A disruption of the correlated neuronal activity associated with the theta rhythmicity is likely to impair these processes (*Uhlhaas and Singer, 2006*; *Geschwind and Levitt, 2007*; *Buzsáki and Watson, 2012*) resulting in atypical social behaviors.

## Materials and methods

### Animals

Sprague-Dawley (SD) male rats (5–6 weeks of age, 250–300 gr) served as subjects while SD or Wistar Hola/Hannover male rats (5–6 weeks of age, 250–300 gr) served as stimuli. All rats were purchased from Harlan Laboratories (Jerusalem, Israel) and housed in groups (2–5 per cage) in the SPF rat facility of the University of Haifa under veterinary supervision, food and water available *ad libidum*, lights on between 7:00 and 19:00. Experiments were performed in a strict accordance with the guidelines of the University of Haifa and approved by its Animal Care and Use Committee.

### Electrodes

We used home-made electrodes for implantation. Stimulating electrodes were prepared by twisting together two stainless steel wires (A-M Systems, Sequim, WA, USA) with bare diameter of 0.005" (Coated-0.008"). Recording electrodes were prepared from Tungsten wire (A-M Systems) with bare diameter of 0.008" (Coated-0.011") soldered to stainless steel wire. For reference/ground wire, we used stainless steel wires attached to a small screw.

### Surgery and electrodes implantation

The rats were anesthetized with subcutaneously injected Ketamine (10% 0.09 cc/100 gr) and Medetomidine (0.1% 0.055 cc/100 gr). Anesthesia level was monitored by testing toe pinch reflexes and held constant throughout surgery with consecutive injections. The body temperature of the rat was kept constant at approximately 37°C, using a closed-loop temperature controller connected to a rectal temperature probe and a heating-pad placed under the rat (FHC, Bowdoin, MA, USA).

Anesthetized rats were fixed in a stereotaxic apparatus (Stoelting, Wood Dale, IL, USA), with the head flat, the skin was gently removed, and holes were drilled in the skull for implantation of electrodes and for reference/ground screw connection. Stimulating electrodes were placed in the left AOB (A/P = +3.0 mm, L/M = +1.0 mm, D/V = −4.0 mm at 50°) and MOB (A/P = +7.08 mm, L/M = +1.0 mm, D/V = −5.5 mm). Recording electrodes were placed in antero-ventral area of the MeA (A/P = −2.4 mm, L/M = +3.18 mm, D/V = −8.5 mm), LS (A/P = −0.24 mm, L/M = +0.4 mm, D/V = −4.4 mm), and Pir (A/P = +3.2 mm, L/M = +3.5 mm, D/V = −5.5 mm), as well as in the NAcc (A/P = +1.2 mm, L/M = +1.4 mm, D/V = −5.8 mm) in later experiments. Each electrode location was verified by its typical field potential signal, evoked in the MeA and LS by AOB stimulation (*Gur et al., 2014*) and in the Pir by MOB stimulation (*Cohen et al., 2013*). Following verification implanted electrodes (one at a time) were fixed by dental cement (Stoelting). When all electrodes were in place, the free ends of the stainless steel wires (including one wire for each stimulation electrode) were wired up to a connector which was then connected to the skull by dental cement, followed by skin is suturing. To avoid a need of soldering, procedure that could damage brain tissue due to excessive heat, we used gold pins inserted to the connector holes under pressure which destroyed the wires isolation to create a contact between the wires and the pins. After surgery, Amoxicillin (15%, 0.07 cc/100 gr) was injected daily (for 3 days) to prevent contamination. Rats allowed recovery for at least 7 days before experiments.

### The experimental setup

All experiments were video-recorded from above the arena (see *Video 1*) by a CCD camera (Prosilica GC1290 GigE, Allied Vision Technology, Taschenweg, Germany). Electrophysiological recordings where made using an 8-channel wireless recording system (W8, Multi Channel Systems, Reutlingen, Germany). Recoded signals (sampled at 1 kHz, low-pass filtered at 0–300Hz) were synchronized with the video recordings by start signal sent through a digital to USB converter (NI USB-6008, National Instruments, Austin, TX, USA) controlled by a self-written LabVIEW program (National Instruments).

The experimental arena comprised a three-layer box (inner dimensions: width—26 cm, length—28 cm, height—40 cm) with door on its front side. The inner layer was made of material (cloth) stretched on cuboid metal carcass to soften mechanical bumps of the recording system. The outer layer was made of adhesive black tape to prevent light entrance. A stainless steel net serves as a faraday cage in between these layers and the Multi-Channel wireless receiver was placed between it and the inner layer. During the

experiment, the arena was illuminated by dim red light. We used a double floor made of two plastic slices that can be separately removed.

## Experiments

Overall, we recorded from 22 animals, of them, 11 were tested with the social paradigm, 6 with the object paradigm (1 animal was tested with both) and 6 animals were tested with both fear conditioning and social encounter. Social recognition memory using anesthetized stimuli was performed in two animals and smell recognition was tested in three animals. The sample size is not always the same for all brain regions since in some of the recorded animals we lost the signals from specific electrodes due to various causes.

At the beginning of each experiment, the tested rat was taken out of its home cage and the wireless transmitter was fastened to the connector on its head by a male-to-male Interconnect header (Mill-Max Mfg. Oyster Bay, NY, USA) with 18 pins. Following 0.5–1 hr of habituation in the experimental arena, the rat was subjected to social, object, smell recognition test (*Figure 2A*), or fear conditioning test (*Figure 7—figure supplement 1*). Each encounter initiated by pressing 'start' button on LabVIEW virtual instrument that sends synchronizing start signal to the camera and the wireless system. Then, during a period of 15 s, the stimulus was removed from its cage and delivered into the experimental arena. At the end of each encounter following stimulus removal, the upper floor slice is taken out and thoroughly cleaned with 70% ethanol and water to remove any odors left by the stimulus. It was then put back below the other slice 5 min after stimulus removal.

## Stimuli

Rat stimuli were individually placed in clean covered plastic box and held in the experiment room throughout the experiment. The two stimulus animals used for each paradigm were always from different rat strains. Anesthetized animal stimuli were subcutaneously injected Ketamine (10% 0.09 cc/100 gr) and Medetomidine (0.1% 0.055 cc/100 gr) 10 min prior to experiment. As object stimuli, we used clean metal office stapler and hole-puncher. For smell recognition, we used small metal-net balls filled with cloth soaked with artificial food smells of citrus and vanilla. The metal-net ball was attached to the cage floor by hot melt adhesive. It should be noted that obviously, both object and smell stimuli are much poorer sources of chemosignals that social stimuli.

## Fear conditioning

Fear conditioning took place in a Plexiglas rodent conditioning chamber with a metal grid floor dimly illuminated by a single house light and enclosed within a sound attenuating chamber (Coulbourn Instruments, Lehigh Valley, PA, USA). Rats were habituated to the chamber for 1 hr before fear conditioning. During fear conditioning rats were presented with five pairings of a tone (CS; 40 s, 5 kHz, 75 dB) that co-terminated with a foot-shock (US; 0.5 s, 1.3 mA). The inter-trial interval was 180 s. The fear recall experiments were conducted a day later in the experimental arena described above, using the same procedure without the electrical foot shocks.

## Histology

After completion of the experiments, the rats were anesthetized and killed with an overdose of Isoflurane (Abbott Laboratories, Chicago, IL, USA). The brains are removed and placed in PFA (4% in PBS) over night, followed by sectioning to 200-μm slices using vibrating slicer (Vibroslice, Campden Instruments, Lafayette, IN, USA). The locations of the implanted electrode tips were identified using binocular and compared to the Pexinos-Watson rat brain atlas (*Paxinos and Watson, 2007*).

## Data analysis

All analyses were done using self-written MATLAB programs (MathWorks, Natick, MA, USA). In all cases, when LFP signals were filtered we used band-pass filter between 5 and 11 Hz (high theta band) using MATLAB 'fir1' function.

### PSD estimation

We estimated Power Spectrum Density (PSD) of LFP signal using multi-tapper approach based on standard Welch's method ('pwelch' function) using 1-s long dpss (discrete prolate spheroidal sequences)

window with 50% overlap. The peak of the PSD curve was considered to be the maximum theta power value for each encounter (*Figure 2*).

## ΔTheta power (ΔTP) calculation
For each brain region, the theta power obtained during Enc. 0 (Base) was subtracted from the TP values of each encounter.

## Spectrogram
For each brain region, spectrograms were computed for each rat per trial using standard 'spectrogram' function with 1-s long dpss window with 50% overlap.

## LFP cross-correlation
We used standard 'xcorr' function with 'coeff' option for cross-correlation between different brain regions of filtered LFP signals for each second. The mean peak cross-correlation value across all 300 s of each encounter was considered to be the cross-correlation value of the encounter (*Figure 4A*).

## Coherence
The coherence between two signals x and y is defined as:

$$Coh_{xy}(f) = \frac{S_{xy}(f)}{\sqrt{S_{xx}(f)S_{yy}(f)}}.$$

We computed the cross-spectrum $S_{xy}(f)$ and the auto-spectra of each signal $S_{xx}(f)$ and $S_{yy}(f)$ using the multitaper method (*Thomson, 1982*), implemented in Chronux 2.0 (*Mitra and Bokil, 2008*), an open-source, data analysis toolbox available at http://chronux.org. Coherograms were computed using a moving window of 2 s shifted in 200 ms increments, 5 tapers, and time-bandwidth of 3. (params.tapers=[TW=3 K=5]; movingwin=[2 0.2];). As spectrograms, coherograms, for each brain region, were computed for each rat per trial. For each brain region, mean coherograms were obtained by averaging coherograms computed per trial across all rats.

## Statistics
Statistical analyses were performed using MATLAB, except for repeated measures ANOVA analyses that were conducted using SPSS (IBM) statistical software. Each brain region was separately analyzed. Parametric t-test and ANOVA tests were used if data were found to be normally distributed (Lilliefors and Shapiro–Wilk tests). Bonferroni's corrections were performed for multiple comparisons using t-test. One-sided t-tests were used when a change in specific direction was expected before the experiment.

## Acknowledgements
We thank Dr Liza Barki-Harrington for a helpful reading of this manuscript. We thank Dr Ido Izhaki and Ms Rotem Gur for generous help with the statistical analyses. This research was supported by the Legacy Heritage Bio-Medical Program of the Israel Science Foundation (grant #1901/08), by the Israel Science Foundation (grant #1350/12) and by a Teva fellowship to AT.

## Additional information

### Funding

| Funder | Grant reference number | Author |
| --- | --- | --- |
| Israel Science Foundation (ISF) | 1901/08, 1350/12 | Shlomo Wagner |
| Teva Pharmaceutical Industries | Graduate student fellowship | Alex Tendler |

The funders had no role in study design, data collection and interpretation, or the decision to submit the work for publication.

### Author contributions
AT, Conception and design, Acquisition of data, Analysis and interpretation of data, Drafting or revising the article; SW, Conception and design, Analysis and interpretation of data, Drafting or revising the article

## Ethics

Animal experimentation: This study was performed in accordance with the recommendations in the Guide for the Care and Use of Laboratory Animals of the National Institutes of Health. All of the animals were handled according to approved institutional animal care and use committee (IACUC) protocols of the University of Haifa. The protocol was approved by the Committee on the Ethics of Animal Experiments of the University of Haifa (Permit Number: 194-10). All surgery was performed under Ketamine and Medetomidine, and every effort was made to minimize suffering.

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
