## [Decision Letter]

[Editors’ note: this article was originally rejected after discussions between the reviewers, but the authors were invited to resubmit after an appeal against the decision.]

Thank you for choosing to send your work entitled “Arousal-driven theta rhythmicity synchronizes neuronal activity in a limbic network during social behavior” for consideration at *eLife*. Your full submission has been evaluated by Eve Marder (Senior editor) and 3 peer reviewers, one of whom is a member of our Board of Reviewing Editors, and the decision was reached after discussions between the reviewers. We regret to inform you that your work will not be considered further for publication. We should stress that it is *eLife's* policy to not ask for extensive revisions, and when it appears that significant work would be needed to bring a manuscript to the stage necessary for publication, we feel it is fairer to the authors to reject the paper and allow them the freedom to prepare it for publication elsewhere, with the advantage of the information found in these reviews.

This manuscript describes an ambitious and interesting study looking at theta oscillatory activity in olfactory, amygdala and basal forebrain areas in rats during social interactions of varying novelty. The authors interpret their results as showing that social memory is correlated with theta activity in the lateral septum and amygdala but not in lower tier olfactory areas. What this means is that theta power in LS and MeA decreases as the same stranger rat is repeatedly presented and then increases when a new rat is presented. Such theta power-tracking of novelty does not occur in AOB or MOB or Pir Ctx or in any area when rats are tested with metal objects or scented objects. In addition there was a theta index which measured the correspondence of MUA with theta power. The authors suggest that the theta response is centrally driven in an arousal dependent manner, and that socially driven theta oscillations in this circuit help bind diverse limbic areas important in this behavior. They further speculate that reported impairment in theta in the autistic spectrum population may contribute to impaired social function.

The reviewers were intrigued but there were several major points that detracted from their enthusiasm:

First and foremost, the social nature of the stimulus was not convincingly parsed from the arousing nature of the stimulus. A non-smelly metal object and an object with a volatile odor that activates MOB glomeruli are not controls for arousal.

Second, there are a number of methodological descriptions (eg theta index was not clearly understood by any of the reviewers) that were unclear and statistical tests which appear to be wrong. A full list is included below.

Third, the authors fail to cite a long literature showing that AOB /MOB theta is a respiratory rhythm which is independent of the olfactory respiratory rhythm. The piriform cortex is known to participate in both the respiratory rhythm and more central theta rhythms, which explains its midpoint position in the coherence strength spectrum among the areas studied here. The large change in AOB and MOB theta power during social stimulation speaks to the odor profile of a conspecific engaging robust sniffing behavior.

We have included the full set of reviews so that you can get a sense of the issues raised by each reviewer in their initial reviews.

*Reviewer #1*:

This paper is an interesting communication that makes a strong case for a physiological correlate for social memory. There are several interesting features to the findings. The social memory is correlated with activity in the lateral septum and amygdala and not in lower tier olfactory areas. While I am not completely convinced by the story, I believe that the data are sufficiently compelling that readers should evaluate the case for themselves.

1) One idea that is not tested is whether or not the social memory is for smell mediated by the VNO. The smell controls were typical volatile odorants that would be processed by the main olfactory epithelium (clove, vanilla). The fact that MOE-mediated odorants do not engage the same responses as individual animals is not that surprising. Observing the response to smells from the urine and feces of two different animals would be more compelling.

2) Theta power is not related to the stimulus animal being awake or anesthetized (but see below) nor is it related to the investigative behavior. This is interpreted as TP being related to neither stimulus nor behavior but rather to an internal state. Another interpretation would be that TP is related to an unrecorded behavior.

3) Conclusions from two animals, two recordings, regarding the issue of awake or anesthetized, is an uncomfortably low sample size. While these observations are worth noting in the Discussion, without more data, I would recommend not making this point part of the results.

4) Finally, how do the results distinguish between the favored top-down model and the possibility of the rhythm as an emergent property of two related areas?

*Reviewer #2*:

This paper addresses the theta rhythm as a coupling mechanism among areas involved in social communication. The data show modulation of theta power and coherence as well as multiunit firing increases and decreases and modulation by theta in response to several types of stimuli. Recordings are done in two primary olfactory areas (MOB, AOB), piriform cortex, medial amygdala and lateral septum. While several of these areas are activated in what is described as a social network, they all participate in many other functions: piriform cortex–olfaction; medial amygdala–emotional processing and olfaction; lateral septum–theta rhythmicity associated with learning and other hippocampal functions. The authors conclude that they have shown that the social areas are preferentially activated by social stimuli. I am not convinced that they have shown anything more than an arousing multimodal stimulus, which happens to be a conspecific and therefore very arousing, can enhance theta rhythms in the limbic areas. The controls are rather impoverished stimuli (clean office supplies, odor in a tea ball). I also have some serious concerns about the statistical methods as detailed below.

Methods and statistics:

1) Were spikes and fields recorded from the same electrodes? If so, was there any check to be sure that the LFP did not contaminate the MUA or vice versa?

2) The band pass for theta is very narrow. This can result in artifactual change in theta power and phase. Please use a much wider band or a low pass filter some larger distance from the end of the band of interest.

3) How do the authors know that the amygdala LFP does not come from the piriform cortex, which is very close by? The spectrograms look quite similar.

4) The method for analyzing MUA is incorrect. From what I can understand of the description (which is not very clear), it appears that all data were pooled across subjects, with those contributing more spikes weighted more, and the resulting pooled data analyzed. Each subject should be analyzed separately and a mean value derived for that subject. Then the statistical tests should be performed.

5) The description of the theta index is very difficult to understand. Is a 60 sec bin correct? That seems a couple of orders of magnitude too large, especially to resolve something on the order of 10s of msecs.

6) It seems that in every analysis a different statistical test is used. Please justify the choice of the various tests. Also, using one-sided t-tests and a large number of t-tests when an ANOVA is a more appropriate test needs to be justified. If many t-tests are used to compare each area separately, a Bonferroni or other correction is necessary to adjust for the multiple tests.

7) deltaTP should use a division of baseline power instead of a subtraction.

8) Figure 3 shows a mean across all brain areas (3A). This measure doesn't offer any information about the system and is an incorrect treatment of power. Power of a signal in any area depends on the circuits in that area. Theta may be big or small but what matters is its power relative to other frequencies in that area.

9) There are many differences between social stimuli and objects that have nothing to do with the social aspect. One is that the live animal stimulus moves in somewhat unpredictable manner or an anesthetized conspecific that is rather confusing. Another is that the odor of the stimulus is very complex, especially as compared to clean office supplies. Also, the waking conspecific can be a threatening stimulus (one aspect of social) and arousal from this emotional state can persist longer than whatever arousal is produced by office supplies. Is the emotional state the object of interest or the social aspect? None of the other stimuli were arousing. Is it possible that the authors have shown that emotional arousal increases theta power in some areas?

10) The authors fail to cite a long literature showing that AOB /MOB theta is a respiratory rhythm which is independent of the olfactory respiratory rhythm. The piriform cortex is known to participate in both the respiratory rhythm and more central theta rhythms, which explains its midpoint position in the coherence strength spectrum among the areas studied here. The large change in AOB and MOB theta power during social stimulation speaks to the odor profile of a conspecific engaging robust sniffing behavior. A scholarly discussion of the results in light of what is already known re respiratory and central theta rhythms is missing.

*Reviewer #3*:

This is a well designed and analyzed experiment describing theta oscillatory activity in olfactory, amygdala and basal forebrain areas in rats during social interactions. The results demonstrate that in response to social interactions with a conspecific, but not with metal objects or scented objects, theta oscillations centering on 8Hz are strongly elevated in all regions, and maintain that activity for at least several minutes after the encounter. Multi-unit activity is also elevated, though only during the encounter. As social interactions habituation over repeated exposure, so does the theta response. The authors suggest that the theta response is centrally driven by in an arousal dependent manner, and that socially driven theta oscillations in this circuit help bind diverse limbic areas important in this behavior. They suggest that reported impairment in theta in the autistic spectrum population may contribute to impaired social function.

Major comments:

1) Despite the authors’ reference to other work, the results would be substantially strengthened by at least one example of simultaneously recorded respiration, definitively showing that theta in this system under these conditions is independent of respiration.

2) Does TP return to baseline prior to the next social stimulus? These data do not seem to be presented. This makes it difficult to interpret TP responses to subsequent social stimuli.

3) There seems to be some disconnect between the various results and how they are interpreted. For example, TP is elevated during social encounters and then stays elevated, while MUA is elevated during social encounters and then returns immediately to baseline. In addition, TI is elevated during social encounters and then is maintained or decreases depending on region. Despite these dramatic differences in time course, the authors seem to imply a single central mechanism based on arousal. This needs to be more fully discussed.

4) The authors suggest that theta oscillations help link large networks, and this may be important in social recognition/cognition. However, the coherence data suggest that while all regions examined express elevated theta, for the most part these oscillations are not coherent across regions. How does this support the authors' hypothesis?

[Editors’ note: what now follows is the decision letter after the authors submitted for further consideration.]

Thank you for sending your work entitled “Distinct types of theta rhythmicity are induced by social and fearful stimuli in a network associated with social memory” for consideration at *eLife*. Your article has been favorably evaluated by Eve Marder (Senior editor), Peggy Mason (Reviewing editor), and one additional reviewer.

The Reviewing editor and the other reviewer discussed their comments before we reached this decision, and the Reviewing editor has assembled the following comments to help you prepare a revised submission.

The addition of the fear conditioning group is a very nice addition to this study and really addresses the arousal issue that so concerned the reviewers in the previous version. There remain matters that we would like you to address in a revision:

1) You have convincingly shown that the effective stimulus has to be either social or it has to be at least more complex than 1 odor (formally a minimum of 2 odors or one odor plus some other cue). Can you acknowledge this possibility? Please give alternate explanations such as an “oligo” mixture of odors full consideration. A measured and sober analysis of potential alternate interpretations will strengthen this paper. So please take this opportunity for revision to produce an even-handed assessment of other possible explanations of the data.

2) The authors may want to consider that the two amygdalae are connected by fibers that pass through the anterior commissure and they are similar biophysically and produce the same sorts of rhythms via the same mechanisms, so this connecting is expected to give rise to strong coherence, despite the distance. Thus strong coherence between the amygdalae is not surprising (despite the distance).

3) In the literature, high frequency theta is assigned to locomotion and sensory activity, while low frequency theta is assigned to immobility or jumping to avoid something (type I and type II theta). In addition to citing the literature concerning low frequency (type II) theta and fear conditioning, could you also cite and discuss the literature demonstrating that high frequency theta accompanies sensory processing and locomotion.

---

## [Author Response]

[Editors’ note: the author responses to the first round of peer review follow.]

We strongly believe that rather than requiring additional experiments, we

can easily address all the concerns raised within a very short time frame. We would therefore greatly appreciate an opportunity to re-submit this manuscript to *eLife*.

The main criticism was for the lack of non-social arousing control. We directly addressed this criticism by examining the theta rhythmicity in the network during fear conditioning, a situation previously shown to be accompanied by arousaldriven theta rhythmicity in other brain regions. These results, presented in the second part of the manuscript, significantly changed the paper that now mainly deals with the LFP signals. Therefore, we omitted the MUA data, the importance of which became marginalized in the revised manuscript.

Reviewer #1:

*This paper is an interesting communication that makes a strong case for a physiological correlate for social memory. There are several interesting features to the findings. The social memory is correlated with activity in the lateral septum and amygdala and not in lower tier olfactory areas. While I am not completely convinced by the story, I believe that the data are sufficiently compelling that readers should evaluate the case for themselves*.

*1) One idea that is not tested is whether or not the social memory is for smell mediated by the VNO. The smell controls were typical volatile odorants that would be processed by the main olfactory epithelium (clove, vanilla). The fact that MOE-mediated odorants do not engage the same responses as individual animals is not that surprising. Observing the response to smells from the urine and feces of two different animals would be more compelling*.

In our manuscript we do not try to tell between the main and accessory olfactory systems in terms of function. Our vision, shared by many other scientists, is that both systems act synergistically to process social chemosensory cues during social interactions and that it would be very difficult to distinguish between their contributions without using genetic tools. Thus, we recorded from areas associated with both systems. Indeed, our results point to the distinction between the olfactory bulb and higher brain centers as more relevant to theta rhythmicity than the difference between the main and accessory systems.

*2) Theta power is not related to the stimulus animal being awake or anesthetized (but see below) nor is it related to the investigative behavior. This is interpreted as TP being related to neither stimulus nor behavior but rather to an internal state. Another interpretation would be that TP is related to an unrecorded behavior*.

The reason why we do not think that any external factor is driving the theta rhythmicity is the variable coherence of this rhythmicity between the various brain regions. This variability cannot be explained if one rhythmic behavior drives all theta rhythmicity in the brain. Moreover, we are continuing a long trail of studies showing that theta rhythmicity in many brain areas is associated with arousal state and that the behavioral parameters such as sniffing and whisking are driven by the internal theta rhythmicity rather than driving it. This issue is widely discussed in the revised manuscript.

*3) Conclusions from two animals, two recordings, regarding the issue of awake or anesthetized is an uncomfortably low sample size. While these observations are worth noting in the Discussion, without more data, I would recommend not making this point part of the results*.

We accept the criticism and omitted the conclusion (regarding the stimulus behavior) from the Results section.

*4) Finally*, *how do the results distinguish between the favored top-down model and the possibility of the rhythm as an emergent property of two related areas?*

The top-down process is supported mainly by the fact that theta rhythmicity is similarly modulated by stimulus novelty in all brain areas, despite the highly variable coherence between them. This suggests that several sources of theta rhythmicity in the brain are influenced by a top-down process which is related to the stimulus novelty. Practically, such influence may be mediated by a common input from a higher brain center or a neuromodulatory system with a brain-wide influence. This issue is discussed in the revised manuscript.

Reviewer #2:

*This paper addresses the theta rhythm as a coupling mechanism among areas involved in social communication. The data show modulation of theta power and coherence as well as multiunit firing increases and decreases and modulation by theta in response to several types of stimuli. Recordings are done in two primary olfactory areas (MOB, AOB), piriform cortex, medial amygdala and lateral septum. While several of these areas are activated in what is described as a social network, they all participate in many other functions: piriform cortex*–*olfaction; medial amygdala*–*emotional processing and olfaction; lateral septum*–*theta rhythmicity associated with learning and other hippocampal functions. The authors conclude that they have shown that the social areas are preferentially activated by social stimuli. I am not convinced that they have shown anything more than an arousing multimodal stimulus, which happens to be a conspecific and therefore very arousing, can enhance theta rhythms in the limbic areas. The controls are rather impoverished stimuli (clean office supplies, odor in a tea ball). I also have some serious concerns about the statistical methods as detailed below*.

*Methods and statistics*:

*1) Were spikes and fields recorded from the same electrodes? If so*, *was there any check to be sure that the LFP did not contaminate the MUA or vice versa?*

We are using methodology which is well accepted in the field and was used by many very well published studies, some of which are cited in the manuscript. Generally, the possibility of contamination of the LFP by the MUA is relevant for high-frequency rhythms such as the gamma rhythm (100Hz), where only one order of magnitude separate between the signal widths. In the case of theta rhythmicity (10Hz), where two order of magnitude, separate between the signals, we do not expect any contamination.

*2) The band pass for theta is very narrow. This can result in artifactual change in theta power and phase. Please use a much wider band or a low pass filter some larger distance from the end of the band of interest*.

The band-pass filter does not cause any distortion of the TP. This can be seen in Figure 2, where unfiltered data is analyzed suing the same methodology and we get the same results. The filtering is only making the peak location clearer.

*3) How do the authors know that the amygdala LFP does not come from the piriform cortex, which is very close by? The spectrograms look quite similar*.

The LFP signals of the MeA and Pir are not especially similar. It can be seen very well in Figure 4—figure supplement 7, where the instantaneous TP of both structures are shown together. Also, the MeA signals show only moderate coherence with the Pir LFP, relative to the LS or contralateral MeA, both of which are remote from the MeA. This issue is discussed in the revised manuscript.

*4) The method for analyzing MUA is incorrect. From what I can understand of the description (which is not very clear), it appears that all data were pooled across subjects, with those contributing more spikes weighted more, and the resulting pooled data analyzed. Each subject should be analyzed separately and a mean value derived for that subject. Then the statistical tests should be performed*.

Not relevant anymore since the MUA data are not included in the revised manuscript.

*5) The description of the theta index is very difficult to understand. Is a 60 sec bin correct? That seems a couple of orders of magnitude too large, especially to resolve something on the order of 10s of msecs*.

Not relevant anymore since the MUA data are not included in the revised manuscript.

*6) It seems that in every analysis a different statistical test is used. Please justify the choice of the various tests. Also, using one-sided t-tests and a large number of t-tests when an ANOVA is a more appropriate test needs to be justified. If many t-tests are used to compare each area separately, a Bonferroni or other correction is necessary to adjust for the multiple tests*.

Generally, each brain region was analyzed separately. All multiple comparisons were corrected using Bonferroni's corrections and this is explicitly written in the relevant figure legends, as well as in a section of “statistics” added to the Methods in the revised manuscript. As explained, one-sided t-tests were used when a change in specific direction was expected before the experiment.

Figure 3: We used repeated measures 1-way ANOVA to examine the possibility of significant differences between encounters and then post-hoc t-tests (corrected for multiple comparisons) to examine the habituation and dishabituation phases. The t-tests were used to minimize the number of post-hoc comparisons within the model.

Figure 4: We used t-test rather than ANOVA since the only desired comparison was between the Encounter and Post periods for each experiment.

Figure 6: t-tests, corrected for multiple comparisons, were used to compare

between the Encounter and Base periods and between the Post and Base periods. We were not interested in comparisons between the encounter and post periods.

Figures 7 and 8: 2-way ANOVA test was used to assess the interaction between the type of experiment (FC or SR) and theta band (low or high) in which changes occur.

*7) deltaTP should use a division of baseline power instead of a subtraction*.

Since the TP is a product of a logarithmic function, the use of subtraction is appropriate. Mathematically, subtraction after log is the same as dividing the parameters before log.

*8)*
Figure 3
*shows a mean across all brain areas (3A). This measure doesn't offer any information about the system and is an incorrect treatment of power. Power of a signal in any area depends on the circuits in that area. Theta may be big or small but what matters is its power relative to other frequencies in that area*.

The mean was removed from the figure.

*9) There are many differences between social stimuli and objects that have nothing to do with the social aspect. One is that the live animal stimulus moves in somewhat unpredictable manner or an anesthetized conspecific that is rather confusing. Another is that the odor of the stimulus is very complex, especially as compared to clean office supplies. Also, the waking conspecific can be a threatening stimulus (one aspect of social) and arousal from this emotional state can persist longer than whatever arousal is produced by office supplies*. *Is the emotional state the object of interest or the social aspect? None of the other stimuli were arousing. Is it possible that the authors have shown that emotional arousal increases theta power in some areas?*

The point with the objects is exactly that they are not arousing and still elicit a similar investigative behavior, as well as habituation and dishabituation, as the social stimuli. Thus, the objects control for the possibility that the theta is not induced by arousal. We now added the fear conditioning experiments, showing that social and fear stimuli, both causing arousal states, yield very different changes in theta rhythmicity and theta coherence in the same network. Thus, the theta rhythmicity differentially reflects distinct arousal states caused by social and fearful stimuli.

*10) The authors fail to cite a long literature showing that AOB /MOB theta is a respiratory rhythm which is independent of the olfactory respiratory rhythm. The piriform cortex is known to participate in both the respiratory rhythm and more central theta rhythms, which explains its midpoint position in the coherence strength spectrum among the areas studied here. The large change in AOB and MOB theta power during social stimulation speaks to the odor profile of a conspecific engaging robust sniffing behavior. A scholarly discussion of the results in light of what is already known re respiratory and central theta rhythms is missing*.

A scholarly discussion was added to the Discussion section, where we address all these points.

Reviewer #3:

*This is a well designed and analyzed experiment describing theta oscillatory activity in olfactory, amygdala and basal forebrain areas in rats during social interactions. The results demonstrate that in response to social interactions with a conspecific, but not with metal objects or scented objects, theta oscillations centering on 8Hz are strongly elevated in all regions, and maintain that activity for at least several minutes after the encounter. Multi-unit activity is also elevated, though only during the encounter. As social interactions habituation over repeated exposure, so does the theta response. The authors suggest that the theta response is centrally driven by in an arousal dependent manner, and that socially driven theta oscillations in this circuit help bind diverse limbic areas important in this behavior. They suggest that reported impairment in theta in the autistic spectrum population may contribute to impaired social function*.

*Major comments*:

*1) Despite the authors’ reference to other work, the results would be substantially strengthened by at least one example of simultaneously recorded respiration, definitively showing that theta in this system under these conditions is independent of respiration*.

We added a detailed discussion regarding the possible involvement of the sniffing cycle in the theta rhythmicity. Briefly, there is no reason to record the sniffing in our case, first because it was previously done with similar paradigms and second because the sniffing cycle itself is known to be modulated by the arousal state and even by expectation to reward. Therefore, it is almost impossible to disconnect the sniffing from the loop. Moreover, the sniffing cycle is probably involved in the entrainment of the OB theta and this may be the reason for the high coherence between the AOB and MOB. However, it cannot be generating the theta of the higher brain centers because of the low coherence between the OB and these centers. This was shown by several papers regarding the hippocampal theta rhythm.

*2) Does TP return to baseline prior to the next social stimulus? These data do not seem to be presented. This makes it difficult to interpret TP responses to subsequent social stimuli*.

These data appear in Figure 4—figure supplement 1, Figure 4—figure supplement 2, Figure 4—figure supplement 3, Figure 4—figure supplement 4, Figure 4—figure supplement 5 and Figure 4—figure supplement 6, where the mean spectrograms of the whole experiment are depicted for each brain region. Generally, this is brain region specific and TP goes back to baseline in the AOB and MOB but not in the other regions, which is another point showing that they are generated by distinct sources of theta rhythmicity.

*3) There seems to be some disconnect between the various results and how they are interpreted. For example, TP is elevated during social encounters and then stays elevated, while MUA is elevated during social encounters and then returns immediately to baseline. In addition, TI is elevated during social encounters and then is maintained or decreases depending on region. Despite these dramatic differences in time course, the authors seem to imply a single central mechanism based on arousal. This needs to be more fully discussed*.

Generally, only the TP is a reflection of the arousal state. In any case, since the MUA data are omitted from the revised manuscript, this is not relevant anymore.

*4) The authors suggest that theta oscillations help link large networks, and this may be important in social recognition/cognition. However, the coherence data suggest that while all regions examined express elevated theta, for the most part these oscillations are not coherent across regions*. *How does this support the authors' hypothesis?*

The hypothesis that brain oscillations in general, and theta rhythmicity in particular, coordinate between remote neural assemblies to promote cognitive processes in not new, and may be the most accepted in the field (see for example [38]). How exactly theta rhythmicity coordinate neural activity is a general and complex issue that was dealt in may review papers, some of which are cited by us (for example [12]) and is out of the scope of this paper. Our coherence data indeed suggest that such coordination is dynamic and may change its pattern in different contexts. In times when the regions recorded by us are not synchronized among themselves, they may be synchronized with other relevant regions. We suggest that such dynamics may be crucial for the enhancement of certain cognitive processes, such as memorization of specific stimuli, in distinct neuronal networks according to their value (social or fearful) and saliency (novel or familiar).

[Editors’ note: the author responses to the re-review follow.]

*The addition of the fear conditioning group is a very nice addition to this study and really addresses the arousal issue that so concerned the reviewers in the previous version. There remain matters that we would like you to address in a revision*:

*1) You have convincingly shown that the effective stimulus has to be either social or it has to be at least more complex than 1 odor (formally a minimum of 2 odors or one odor plus some other cue). Can you acknowledge this possibility? Please give alternate explanations such as an ‘”oligo” mixture of odors full consideration. A measured and sober analysis of potential alternate interpretations will strengthen this paper. So please take this opportunity for revision to produce an even-handed assessment of other possible explanations of the data*.

We added a paragraph dealing with this issue to the Discussion section (second paragraph).

*2) The authors may want to consider that the two amygdalae are connected by fibers that pass through the anterior commissure and they are similar biophysically and produce the same sorts of rhythms via the same mechanisms, so this connecting is expected to give rise to strong coherence, despite the distance. Thus strong coherence between the amygdalae is not surprising (despite the distance)*.

The lack of a direct synaptic connection between the MeAs of the two hemispheres is well documented, especially in the comprehensive work of Canteras, Simerly and Swanson (The Journal of Comparative Neurology 360:213-245, 1995, see paragraphs regarding contralateral projections at pp. 225, 227: “Virtually no anterograde labeling was observed in the contralateral amygdala”) specifically for rats, and is also evident in the Mouse Connectivity section of the Allen Brain Atlas. We added a reference to the work of Canteras et al. in the relevant place towards the end of the Discussion section.

*3) In the literature, high frequency theta is assigned to locomotion and sensory activity, while low frequency theta is assigned to immobility or jumping to avoid something (type I and type II theta). In addition to citing the literature concerning low frequency (type II) theta and fear conditioning, could you also cite and discuss the literature demonstrating that high frequency theta accompanies sensory processing and locomotion*.

We added a full discussion of the two types of theta recorded in the hippocampus and the relationship of them to other theta rhythms recorded in the brain to the Discussion section (third paragraph).